# Immune Modulation and Efficacy of Tixagevimab/Cilgavimab Pre-Exposure Prophylaxis in Lung Transplant Recipients During the Omicron Wave

**DOI:** 10.3390/ijms26083696

**Published:** 2025-04-14

**Authors:** Lolita Sasset, Roberta Angioni, Nicolò Presa, Ricardo Sánchez-Rodríguez, Claudia Cozzolino, Nicole Bertoldi, Serena Marinello, Monica Loy, Maria Mazzitelli, Federico Rea, Annamaria Cattelan, Barbara Molon

**Affiliations:** 1Infectious and Tropical Diseases Unit, University Hospital of Padua, 35128 Padua, Italy; lolita.sasset@aopd.veneto.it (L.S.); annamaria.cattelan@aopd.veneto.it (A.C.); 2Department of Biomedical Sciences, University of Padua, 35131 Padua, Italy; ricardo.sanchezrodriguez@unipd.it (R.S.-R.); barbara.molon@unipd.it (B.M.); 3Fondazione Istituto di Ricerca Pediatrica-Città della Speranza, 35127 Padua, Italy; 4Department of Cardiac, Thoracic, Vascular Sciences, and Public Health, University Hospital of Padua, 35128 Padua, Italy; claudia.cozzolino@studenti.unipd.it (C.C.);; 5Department of Molecular Medicine, University of Padua, 35131 Padua, Italy

**Keywords:** lung transplant recipients, SARS-CoV-2, immune response, tixagevimab/cilgavimab

## Abstract

Lung transplant recipients are at increased risk of severe COVID-19 due to lifelong immunosuppressive therapy, which impairs both innate and adaptive immune responses. Identifying effective supportive therapies is essential for mitigating the heightened vulnerability of this population. This study investigated the effects of tixagevimab/cilgavimab, a monoclonal antibody therapy, as pre-exposure prophylaxis (PrEP) in this population. A prospective study was conducted on 19 lung transplant recipients at Padua University Hospital, Italy, during the Omicron variant wave (May–June 2022). Participants received tixagevimab/cilgavimab intramuscularly and were monitored for 180 days. SARS-CoV-2-specific antibody levels were measured at baseline (T0), one month (T1), and three months (T3) post-treatment. Cytokine profiles and clinical outcomes, including SARS-CoV-2 infections, were also assessed. At baseline, 50% of patients had negative antibody responses, but one-month post-treatment, all patients exceeded 700 kBAU/mL (median 3870 kBAU/mL), with levels decreasing but remaining positive at three months (median 1670 kBAU/mL). Remarkably, a higher level of circulating IL-18 was found at T3 in comparison to T0 in patients who did not experience COVID-19 after PrEP. This finding aligns with IL-18’s primary role in stimulating type-1 T helper (Th1) cell responses, necessary for the induction of virus-specific cytotoxic T lymphocytes (CTLs). These results suggest that tixagevimab/cilgavimab may induce a systemic immune signature that could contribute to priming the immune response against SARS-CoV-2, potentially mediated by interactions with immune cell subsets.

## 1. Introduction

Lung transplant recipients face significant challenges with SARS-CoV-2 infection, being at a higher risk for severe disease, hospitalization, and mortality compared to the general population [1]. Pre-existing lung conditions, the presence of comorbidities, and the effects of immunosuppressive medications may all contribute to reduce both innate and adaptive immune responses in this vulnerable population [2]. Research indicates that many lung transplant recipients demonstrate suboptimal antibody responses following SARS-CoV-2 infection or vaccination [3], underscoring the need for more therapeutic options to safeguard this high-risk population. Tixagevimab/cilgavimab, a combination of two long-acting monoclonal antibodies, has been developed as a pre-exposure prophylaxis against COVID-19. Tixagevimab/cilgavimab targets the spike protein of SARS-CoV-2, preventing the virus from entering human cells. Clinical studies have demonstrated that this combination of monoclonal antibodies significantly reduces the risk of symptomatic COVID-19 in solid organ transplant recipients, including lung transplant recipients, showing promising results in preventing COVID-19 infection, providing an effective prevention of COVID-19, especially in cases where vaccine-induced immunity is insufficient [4].

The immune response to SARS-CoV-2 involves both innate and adaptive components. Initially, the innate immune system detects viral presence through pattern recognition receptors (PRRs), leading to the activation of inflammatory pathways and the production of cytokines and interferons (IFNs). This early response is critical for controlling viral replication and activating adaptive immunity. A coordinated adaptive immune response is characterized by the activation of CD4+ T helper cells, which assist in antibody production by B cells, and CD8+ cytotoxic T cells that target and eliminate infected cells. Antibody responses, particularly neutralizing antibodies against the spike protein, play a key role in preventing viral entry into host cells [5].

Additionally, clear evidence indicates that the circulating cytokine milieu affects the immune response to SARS-CoV-2 infection, thus representing a crucial determinant for COVID-19 clinical outcomes [6]. In immunocompromised patients, the immune landscape is significantly altered, characterized by deficiencies in both innate and adaptive immunity. These individuals often exhibit impaired T and B cell responses, reduced cytotoxic activity of NK cells, and a diminished ability to produce neutralizing antibodies. From a cytokine perspective, immunosuppressed patients may fail to mount an effective pro-inflammatory response to infection, resulting in delayed viral clearance and prolonged viral replication. This dysregulated immune and cytokine landscape underscores the need for tailored therapeutic approaches, including preventive interventions, to mitigate the risk of severe COVID-19 outcomes in this vulnerable population. Indeed, the relevance of the systemic cytokine landscape in immunocompromised patients undergoing pre-exposure prophylaxis (PrEP) for SARS-CoV-2 has not been yet investigated and data on lung transplant recipients are quite limited [7,8]. To address this point, we assess and monitor over time the immune response and the cytokine dynamics among lung transplant’s patients receiving tixagevimab/cilgavimab as PrEP.

## 2. Results

Overall, 19 lung transplant recipients were included in the analysis and followed throughout this study for 180 days after receiving PrEP (detailed characteristics are depicted in Table 1). The group was fairly balanced in terms of gender, with males comprising 47% and females 53%. The median age at the first visit was 51 years, and the median time since transplantation was 5 years. In detail, among the 19 patients, 8 had been receiving immunosuppressive therapy for more than 5 years (range: 6–15 years), 8 for 2–5 years, and 3 patients had been on immunosuppressive therapy for less than 1 year. All patients had undergone bilateral lung transplants. The main underlying lung pathologies were cystic fibrosis/bronchiectasis (47%) and fibrosis/interstitial lung disease (27%). Most patients were on dual immunosuppressive therapy (79%), with cyclosporine (53%) and mycophenolate (47%) being the two most commonly used agents. Although nearly all patients had completed a full course of vaccination for SARS-CoV-2 (95%), at baseline (T0), 50% exhibited a negative humoral immune response (antibody binding units kBAU <50/mL), while 42% had a low response (kBAU between 50 and 700/mL). Two patients (8%) demonstrated an adequate immunological response with kBAU exceeding 700/mL. All 19 patients received tixagevimab/cilgavimab as bilateral intramuscular injections of 150 mg each (single dose).

At months 1 and 3 of follow-up, all 19 patients exhibited a positive level of SARS-CoV-2 antibody binding units (kBAU > 700/mL), with a median value of 3870 kBAU at month 1 (T1) and 1670 kBAU at month 3 (T3) of follow-up. A significant reduction in the level of kBAU was observed at T3 (Figure 1A). Notably, the reduction was significantly higher in patients older than 40 years (Figure 1B). During the study period, a total of 6 patients (32%) developed a SARS-CoV-2 infection. Among these individuals, we did not observe a significant increase in the Antibody (Ab) titers at the different time points. Conversely, we detected a significant trend in patients who did not experience COVID-19 after PrEP (Figure 1C). However, when considering the Ab level at T1 and T3, there was no significant difference between the two groups (Figure 1D).

Importantly, a previous SARS-CoV2 infection did not influence the humoral response in immunosuppressed patients (Figure 1E). Patients who were treated with immunosuppressive drugs for less than one year and those treated for 2–5 years did not exhibit a significant difference in Ab titers at various time points. In contrast, individuals who underwent immunosuppressive therapy for more than five years demonstrated significant differences in Ab titers at the three distinct time points which may be attributed to their lower baseline level of kBAU at T0 compared to other patients. Five patients (83%) developed a COVID-19, classified as mild, and four of them (67%) received early treatment for SARS-CoV2 infection, which included oral antiviral or a combination of other monoclonal antibodies. One patient (17%), who did not receive early treatment, experienced a severe infection that required hospitalization, supportive oxygen therapy and steroid treatment, ultimately resulting in a positive outcome. No deaths from COVID-19 or other conditions were recorded during the follow-up in this cohort of patients.

The cytokine profile of our patients, which measures the circulating plasma levels of 48 cytokines within our cohort, is detailed in Figure 2A. The heatmap displays standardized values (z-scores) for each biomarker across the entire sample, with transplant recipients showing above-average values marked in red and those with below-average values in blue. Overall, the plot shows a pattern where a stronger response in certain biomarkers correlated with above-average values in others. This trend is particularly evident in patients 4, 5, 9, and 16.

Among the analyzed cytokines, we pointed out that IL-18 was the only soluble factor significantly increasing at T3 (Figure 2B). At this time point (T3), IL-18 upregulation was independent from patient age (Figure 2C). A significant increase in this cytokine over time was observed in patients undergoing an immunosuppressive regimen for more than 5 years (Figure 2D). Remarkably, a higher level of circulating IL-18 was found at T3 in comparison to T0 in patients who did not experience COVID-19 after PrEP (Figure 2E). However, at the specified time points, we were unable to identify any significant differences in IL-18 levels between patients who had a subsequent SARS-CoV-2 infection and those who did not (Figure 2F).

When we analyzed the full cytokine profile of patients who experienced a subsequent SARS-CoV-2 infection in comparison to those who tested negative at T0, T1, and T3, the heatmap revealed no significant differences in the inflammatory profile, with relative percentage changes ranging from −75% to 75% (Figure 3A). However, at T3, patients with SARS-CoV-2 infection generally exhibited lower levels (indicated by blue) of all the cytokines analyzed. Additionally, when examining the differences in delta values between time points across the groups, there was a more pronounced increase in the inflammatory response among patients who developed the infection compared to those who did not, as reflected in red.

Our analysis indicated a similar trend for IL-18, CCL22 (Figure 3B,C), as well as TNF-alpha (Figure 3D). However, TNF-alpha levels were lower in patients who experienced a subsequent SARS-CoV-2 infection compared to those who did not with a statistically significant decrease at T3 (Figure 3E).

The cytokine MIG exhibited a different pattern, as it did not demonstrate time-dependent regulation like the previously mentioned cytokines. However, it did show a significant decrease between the two groups at all analyzed time points (Figure 3F,G).

## 3. Discussion

This study provides valuable insights into the immune responses of lung transplant recipients following the administration of PrEP with tixagevimab/cilgavimab during the Omicron wave of the COVID-19 pandemic. These patients are particularly vulnerable to severe COVID-19 due to their immunocompromised status, which can result from both the underlying lung disease and the immunosuppressive therapies required to prevent organ rejection. Previous studies have highlighted the benefits of tixagevimab/cilgavimab to lung transplant recipients, showing its efficacy in protecting against symptomatic SARS-CoV-2 infections and reducing severe outcomes [8,9].

Although our cohort is small, it is comparable to those reported in the literature in terms of time since transplantation, immunosuppressive regimens, and underlying conditions [8]. Adverse events associated with monoclonal antibodies, such as severe hypersensitivity reactions (including anaphylaxis), cardiovascular complications, and injection-related reactions, have been documented in previous studies like PROVENT [10]. However, none of these adverse effects—including coronary or thromboembolic events, chills, or injection site discomfort—were observed during clinical follow-up in our study population.

Our findings also highlight the importance of proactive management in transplant recipients. Indeed, the primary aim of our research was to analyze the dynamics of both humoral responses and cytokine profiles in this vulnerable population following the administration of monoclonal antibodies, identifying potential biomarkers of immune readiness. Despite a high vaccination rate (95%), a substantial portion of patients exhibited suboptimal humoral responses at baseline, highlighting the challenges faced by immunosuppressed individuals in achieving adequate immunity against SARS-CoV-2. The results demonstrated a robust increase in SARS-CoV-2 antibody binding units (kBAU) at both one month (T1) and three months (T3) post-treatment, with median values significantly higher than baseline. While we did not perform neutralization assays, we are aware that several studies have indicated that anti-SARS-CoV-2 receptor binding domain (RBD)-antibodies closely correlate with neutralizing antibody levels in COVID-19 patients [11,12,13]. Our results align with previous studies suggesting that the administration of PrEP effectively enhanced the humoral immune response in these immunosuppressed patients (1,2). Of note, despite the decline at T3, antibody titers remained above the protective threshold, suggesting that a single dose of tixagevimab/cilgavimab provides sustained humoral protection over the 3-month follow-up period. However, some concerns arise among individuals over 40 years of age and individuals with a long-lasting immunosuppression (>5 years) in which a significant reduction in antibody levels was observed at T3. This decline may be attributed either to the effect of immune aging or to the effects of prolonged immunosuppressive therapy, which can impair the ability to maintain stable antibody levels over time.

Interestingly, while 32% of our patients developed SARS-CoV-2 infections during this study, we did not observe a significant increase in antibody titers at various time points for this group. This finding suggests that the presence of a subsequent infection does not necessarily correlate with an enhanced humoral response in immunosuppressed patients, contrasting with the trend seen in patients who did not contract COVID-19. Moreover, the lack of significant differences in antibody levels between those with prior infections and those without suggests that humoral immunity alone may not be sufficient to fully mitigate infection risk in this population. Instead, these findings emphasize the importance of integrated immune responses, including cytokine-mediated effects. The systemic cytokine milieu appears to play a pivotal role in shaping immune outcomes post-PrEP. Among the analyzed cytokines, IL-18 emerged as a significant factor, showing a time-dependent increase, particularly in patients who did not experience COVID-19. This aligns with its established role in promoting Th1 responses and IFN-γ production, which are essential for CTL activation and effective viral clearance [14].

Notably, the increase in IL-18 was more pronounced in patients undergoing immunosuppressive therapy for over 5 years, suggesting a potential adaptation or modulation of the immune system over prolonged immunosuppression. This could be also related to the different immunosuppressive treatment (schedule and drugs) between the two patient groups. Notably, although IL-18 was the only cytokine significantly increased at T3 in patients who remained uninfected, we acknowledge that this association does not imply causation. The observed elevation may reflect an indirect effect of tixagevimab/cilgavimab on systemic immune activation rather than a direct protective mechanism. IL-18 plays a dual role in immunity. In the presence of IL-12, IL-18 promotes antiviral Th1 responses; however, in its absence, IL-18 can drive Th2-skewed responses via IL-2, as documented in prior studies [15]. This duality may partly explain why IL-18 alone did not differentiate between patients who developed COVID-19 and those who did not at specific time points. In addition to IL-18, we identified CCL22 and TNF-α as cytokines with a time-dependent increase post-PrEP. CCL22, associated with T-cell recruitment, may enhance local immune responses, while TNF-α, a key pro-inflammatory cytokine, plays a crucial role in orchestrating antiviral immunity. The significant reduction in TNF-α levels in the COVID-19 group, particularly at T3, suggests a blunted inflammatory response in these patients, which may impair viral clearance and contribute to infection susceptibility. Conversely, MIG (CXCL9), which showed consistently lower levels in the COVID-19 group, highlights potential deficits in T-cell chemotaxis and immune activation [16]. The heatmap analysis further supports a broad modulation of cytokine responses by PrEP, with notable increases in inflammatory mediators over time. The cytokine increase may be suggestive of a putative indirect mechanism by which tixagevimab/cilgavimab primes the immune system, complementing its direct neutralizing activity against SARS-CoV-2. The mild severity of COVID-19 in patients who developed breakthrough infections, with only one case requiring hospitalization, emphasizes the protective effect of PrEP. The ability of tixagevimab/cilgavimab to mitigate severe outcomes is consistent with its role in neutralizing viral entry and reducing viral load. The findings also suggest that early antiviral intervention can further attenuate disease severity, highlighting the importance of proactive management in transplant recipients.

Overall, our data highlight a dual protective effect of tixagevimab/cilgavimab in lung transplant patients, mediated through an active humoral response and cytokine modulation. These findings contribute to the growing evidence supporting tailored immunoprophylaxis in immunocompromised populations and underscore the need for further research to refine these strategies.

## 4. Materials and Methods

### 4.1. Participants, Study Design and Data Collection

A prospective interventional study was conducted from May to June 2023, involving lung transplant recipients under active monitoring at the Transplant Center of the University Hospital of Padova. All eligible patients were contacted via telephone and invited to participate in this study. Inclusion criteria required participants to be able to visit the transplant center during the COVID-19 pandemic, have no active or opportunistic infections, consent to undergoing blood tests as part of their follow-up, and provide written informed consent. Exclusion criteria included refusal of intramuscular injections, a body weight below 40 kg, a recent SARS-CoV-2 infection (within the last three months), or coagulation disorders preventing the safe discontinuation of anticoagulant therapy. In total, 19 patients met the eligibility criteria and were enrolled. Each participant was clinically assessed for six months following the administration of tixagevimab/cilgavimab. This study officially concluded on 30 January 2023, coinciding with the completion of the six-month follow-up for the last enrolled patient. A graphical summary is provided in Figure 4.

Tixagevimab/cilgavimab was administered by intramuscular injections at the dose of 300 mg according to clinical indication between May and June 2022, when Omicron SARS-CoV-2 variant was prevalent in Italy. We assessed the anti-SARS-CoV-2 binding antibody titers (kBAU/mL) by using a commercially available immunoassay, able to detect anti- SARS-CoV-2 receptor binding domain (RBD) IgG (Snibe Diagnostics, New Industries Biomedical Engineering Co., Ltd. [Snibe], Shenzhen, China). All analyses were conducted using the MAGLUMI™2000Plus (Snibe Diagnostics), with results reported in kBAU/mL. The immunological response to prior SARS-CoV-2 vaccination and following PrEP was categorized as negative/low titers when antibody binding units (kBAU) were less than 50/mL, mid titers for kBAU between 50 and 700/mL, and high titers for kBAU greater than 700/mL, according with the WHO International Standard for COVID-19 immunoglobulin [17]. At baseline, demographic and clinical information, including data on underlying lung disease, the date of transplantation, lung transplant procedure, type of immunosuppressive regimen, comorbidities prior SARS-CoV-2 vaccination, type of vaccine used, ongoing immunosuppressive therapy, and any previous SARS-CoV-2 infections were collected. Patients were followed up for 180 days after PrEP administration. Blood samples for the detection of anti-Spike antibody titer and immunological study were collected at baseline (before monoclonal antibody administration), and at month 1 and 3 of follow-up. To analyze the impact of the immunosuppressive regimen on the antibody response within our patient cohort we categorized the patients into 3 groups: (i) Patients undergoing an immunosuppressive regimen for less than one year (ii) patients treated for one to five years, (iii) patients treated for more than five years.

Throughout the follow-up period, a new SARS-CoV-2 infection was identified by any positive polymerase chain reaction (PCR) test result. The severity of the infection was categorized based on the National Institutes of Health COVID-19 treatment guidelines, which classify cases as asymptomatic, mild, moderate, severe, or critical [18].

In addition, data were collected regarding the early administration of secondary prophylaxis therapy in the event of a new infection, the hospitalization rate, and the outcomes of the patients

### 4.2. Luminex and ELISA Assay

Peripheral blood from enrolled lung transplant’s patients was collected in EDTA tubes and stored at 4 °C prior to processing. Plasma was isolated by Ficoll procedure and stored at −80 °C until the analysis. A total of 48 analytes were measured by multiplex biomarker assays, based on Luminex xMAP technology (Merck Millipore, Burlington, MA, USA) following manufacturer’s instructions. Blood specimens were collected at 3 different time points: at the baseline, before drug administration (T0), after 1 month (T1) and after 3 months (T2).

### 4.3. Statistical Analysis

Descriptive statistics for the cohort were computed, representing categorical variables as frequencies and percentages, and continuous variables as means with standard deviations or medians with interquartile ranges (IQR), as appropriate. Bivariate analyses were conducted to assess differences in clinical features and biomarkers based on age at the time of treatment (≤40 vs. >40 years), duration of immunosuppressive therapy (1 year, 2–5 years, >5 years), antibody response (kBAU/mL at T0 < 50, 50–700, >700), and history of prior or subsequent SARS-CoV-2 infection (yes vs. no). Statistical tests included Fisher’s exact test or, when applicable, Pearson’s Chi-squared test for categorical variables; the two-sample t-test or the Wilcoxon rank sum test, one-way analysis of variance (ANOVA), or the Kruskal–Wallis rank sum test for continuous variables depending on the distribution of data and the number of grouping classes. Differences in antibody response and biomarkers over time (T0, T1, and T3) were evaluated using the Friedman test, a non-parametric method for one-way repeated-measures ANOVA. Additionally, two-way mixed ANOVAs were performed to examine the combined effects of time and grouping variables such as age, duration of immunosuppressive therapy, antibody response, and history of SARS-CoV-2 infection (prior or subsequent to treatment). For both analyses, pairwise comparisons were conducted using the Wilcoxon rank sum test. The Bonferroni correction was applied to adjust *p*-values separately for each cytokine comparison to minimize the reduction in statistical power given the small sample size.

The distribution of biomarkers across subjects and time points for the entire cohort of transplant recipients was visualized using heatmaps, with values standardized for each biomarker (z-scores). Heatmaps were also used to display pairwise differences in biomarker between patient subgroups, representing the percentage relative change in median values (between first group vs. second one). This visualization was performed for individual time points and for absolute deltas between time points (T1–T0, T3–T1, T3–T0) to investigate whether temporal changes in biomarkers differed between subjects with distinct characteristics, such as age, duration of immunosuppressive therapy, antibody response, and SARS-CoV-2 infection history. A *p*-value <0.05 was considered statistically significant. All data manipulations, analyses, and visualizations were performed using Python 3.8.18 and R 4.2.2.

## 5. Limitations and Future Directions

While this study provides valuable insights, several limitations must be acknowledged. The small sample size limits the impact of our findings, and the absence of a randomized control group precludes definitive conclusions about PrEP efficacy compared to other prophylactic strategies. Additionally, the lack of cellular immune cell profiling, such as T-cell or NK cell activity, leaves gaps in understanding the full spectrum of immune responses in our patient cohort. Future studies should explore the long-term durability of PrEP-induced immunity and its interactions with evolving SARS-CoV-2 variants. Expanding cytokine analyses to include functional assays could provide deeper insights into the interplay between humoral and cellular immunity. Moreover, investigating combination strategies, such as integrating PrEP with tailored vaccination schedules, may optimize protective outcomes in transplant recipients.

## Figures and Tables

**Figure 1 ijms-26-03696-f001:**
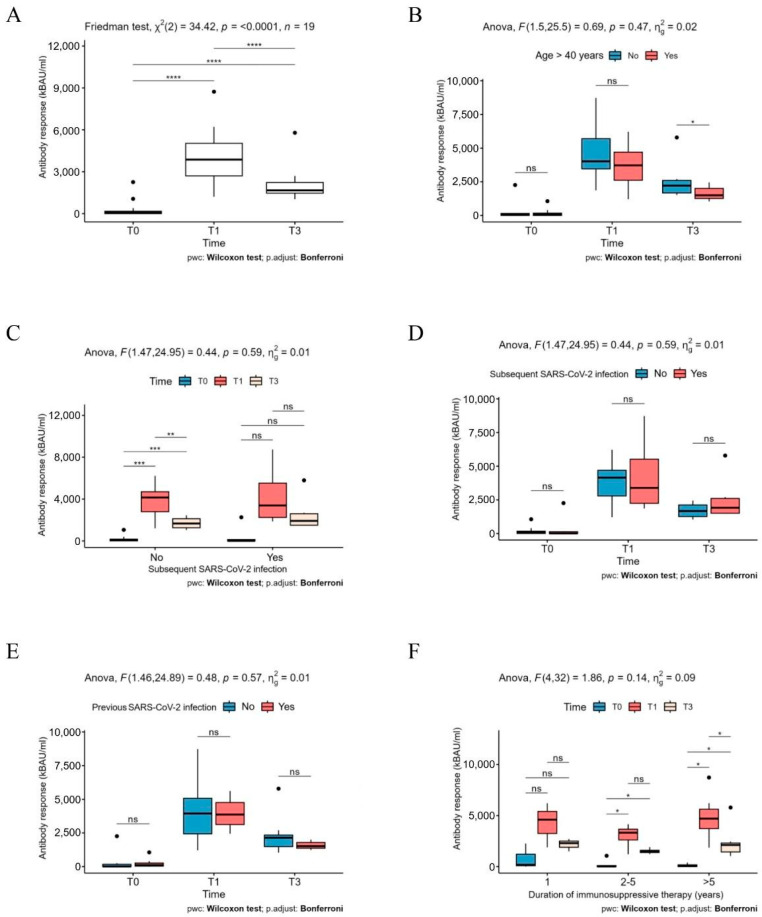
Humoral immune response in the patient cohort. (**A**) Serum titer of anti-SARS-CoV-2 antibodies (kBAU/mL) in patients at the indicated time points (T0, T1, T3); (**B**) serum titer of anti-SARS-CoV-2 antibodies (kBAU/mL) in patients: (**B**) aged ≤40 years-old or >40 years old; (**C**,**D**) who experience or not a subsequent SARS-CoV-2 infection at the indicated time points (T0, T1, T3); (**E**) who experienced or not a previous SARS-CoV-2 infection at the indicated time points (T0, T1, T3); (**F**) undergoing immunosuppressive regimen for 1, or 2–5 or >5 years at the indicated time points (T0, T1, T3). One-way repeated-measures non-parametric ANOVA (**A**), the Friedman test, and two-way mixed ANOVA (**B**–**F**) were conducted to evaluate the effects of time and grouping variables. Pairwise comparisons were carried out using the Wilcoxon rank sum test with *p*-values adjusted using the Bonferroni method. Significance levels were defined as follows: ns (not significant) *p* ≥ 0.05; * *p* < 0.05; ** *p* < 0.01; *** *p* < 0.001; **** *p* < 0.0001.

**Figure 2 ijms-26-03696-f002:**
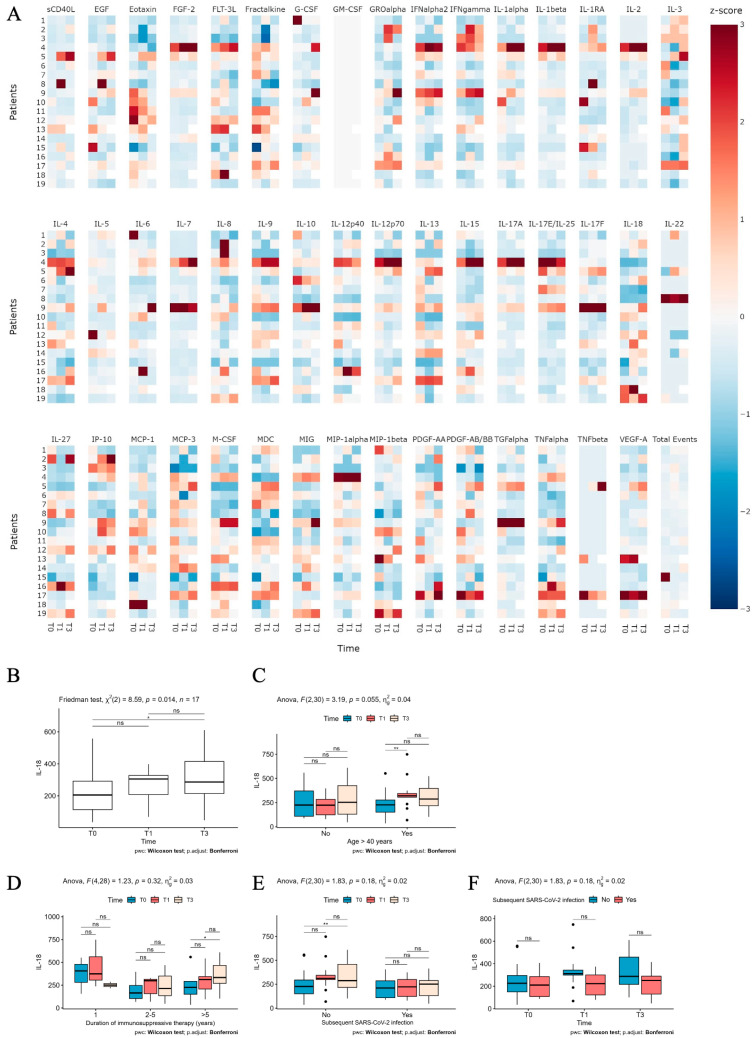
Circulating immune factors in the patient cohort. (**A**) Heatmap of 48 serum analytes in patients (rows) at the indicated time points (T0, T1, T3) (columns). Colors represent the standardized values (z-scores) for each biomarker, where white indicates values close to the overall mean, while red and blue represent values, respectively, higher and lower than the mean. (**B**) Serum level of circulating IL-8 (pg/mL) at the indicated time points (T0, T1, T3). Serum level of circulating IL-8 (pg/mL) in patients: (**C**) aged ≤40 years old or >40 years old; (**D**) undergoing immunosuppressive regimen for 1, or 2–5 or >5 years at the indicated time points (T0, T1, T3); (**E**,**F**) who experience or not a subsequent SARS-CoV-2 infection at the indicated time points (T0, T1, T3). One-way repeated-measures non-parametric ANOVA (**B**), the Friedman test, and two-way mixed ANOVA (**C**–**F**) were conducted to evaluate the effects of time and grouping variables. Pairwise comparisons were carried out using the Wilcoxon rank sum test with *p*-values adjusted using the Bonferroni method. Significance levels were defined as follows: ns (not significant) *p* ≥ 0.05; * *p* < 0.05; ** *p* < 0.01.

**Figure 3 ijms-26-03696-f003:**
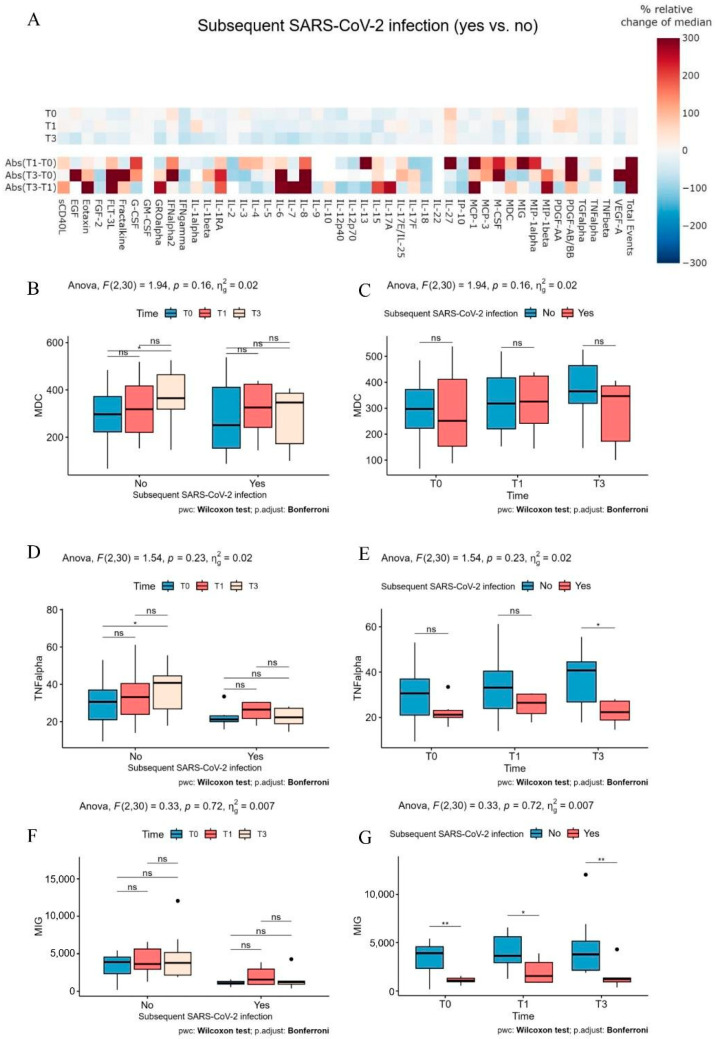
CCL22/MDC, TNF-alpha and MIG levels in patients. (**A**) Heatmap of pairwise differences in 48 serum analytes (rows) at individual time points and absolute deltas between time points (T1–T0; T3–T1; T3–T0) (columns) comparing patients who experienced a subsequent SARS-CoV-2 infection to those who did not. Differences are shown as the percentage relative change in median values. White indicates minimal differences between the two groups at a given time point or time delta and analyte, while red and blue represent relatively higher or lower median values, respectively. (**B**,**C**) Serum level of circulating CCL22/MDC (pg/mL)at the indicated time points (T0, T1, T3) in patients who experience or not a subsequent Sars-CoV-2 infection. (**D**,**E**) Serum level of circulating TNF-alpha (pg/mL) at the indicated time points (T0, T1, T3) in patients who experience or not a subsequent Sars-CoV-2 infection. (**F**,**G**) Serum level of circulating MIG (pg/mL) at the indicated time points (T0, T1, T3) in patients who experience or not a subsequent Sars-CoV-2 infection. Two-way mixed ANOVA (**B**–**G**) were conducted to evaluate the effects of time and grouping variables. Pairwise comparisons were carried out using the Wilcoxon rank sum test with *p*-values adjusted using the Bonferroni method. Significance levels were defined as follows: ns (not significant) *p* ≥ 0.05; * *p* < 0.05; ** *p* < 0.01.

**Figure 4 ijms-26-03696-f004:**
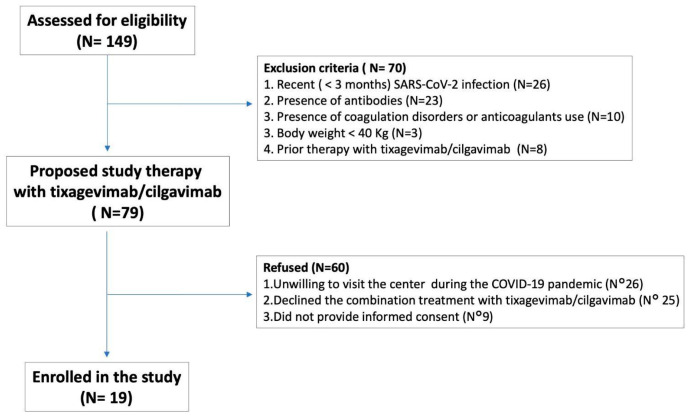
Study flowchart of included patients.

**Table 1 ijms-26-03696-t001:** Lung transplant patients cohort descriptive variables.

Variable	N = 19
Sex, *n* (%)	
Male	9 (47)
Female	10 (53)
Age at visit, median years (IQR)	51 (40–64)
Time after transplant, median years (IQR)	5 (6 months–15 years)
Lung transplant procedure, *n* (%)	
Bilateral	19 (100)
Unilateral	0 (0)
Underlying disease, *n* (%)	
Cystic Fibrosis/Bronchiectasis	9 (47)
Fibrosis/interstitial lung disease	5 (27)
COPD/Emphysema	4 (21)
Pulmonary vascular disease	1 (5)
Immunosuppression, *n* (%)	
Cyclosporine	10 (53)
Mycophenolate	9 (47)
Everolimus	8 (42)
Tacrolimus	6 (32)
Dual Immunosuppression, *n* (%)	15 (79)
Pre-study SARS-CoV-2 infection *n* (%)	7 (37)
Vaccination status, *n* (%)	
No or incomplete vaccination	1 (5)
Full vaccination (at least 3 doses)	18 (95)
Humoral immune response after vaccine, *n* (%)	
<50 kBAU/ml	9 (50)
50–700 kBAU/ml	8 (42)
>700 kBAU/ml	2 (8)
Follow-up, median days after first visit during study (IQR)	180 (172–194)
Humoral immune response 1 month after PrEP, *n* (%)	
<50 kBAU/ml	0 (0)
50–700 kBAU/ml	0 (0)
>700 kBAU/ml	19 (100)
Humoral immune response 3 month after PrEP, *n* (%)	
<50 kBAU/ml	0 (0)
50–700 kBAU/ml	0 (0)
>700 kBAU/ml	19 (100)
SARS-CoV-2 infection after tixagevimab/cilgavimab, *n* (%)	6 (32)
COVID-19 severity during study period, *n* (%)	
Asymptomatic	0 (0)
Mild	5 (83)
Moderate	0 (0)
Severe	1 (17)
Critical	0 (0)
Early treatment of SARS-CoV-2 infection, *n* (%)	
None/reduction in immunosuppression	2 (33)
Remdesivir	1 (17)
Molnupiravir	2 (33)
Sotrovimab	1 (17)
Hospitalization, *n* (%)	1 (17)
Death, *n* (%)	0 (0)
COVID-19-associated death, *n* (%)	0 (0)

## Data Availability

The original contributions presented in this study are included in this article. Further inquiries can be directed to the corresponding author.

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
