# Peer review of "Immune Modulation and Efficacy of Tixagevimab/Cilgavimab Pre-Exposure Prophylaxis in Lung Transplant Recipients During the Omicron Wave"

_ijms, 2025, doi:10.3390/ijms26083696_

Round 1
Reviewer 1 Report
Comments and Suggestions for Authors
The manuscript presented for review titled “Immune Modulation and Efficacy of Tixagevimab/Cilgavimab Pre-Exposure Prophylaxis in Lung Transplant Recipients During the Omicron Wave" presents a study on the effects of tixagevimab/cilgavimab (T/C) pre-exposure prophylaxis (PrEP) in lung transplant recipients during the Omicron wave.
The subject matter of this article is important as it provides valuable clinical and immunological insights, particularly regarding cytokine responses and antibody dynamics. The systematic analysis of the topics raised by the Authors has been presented in a clear and coherent manner. The language of the work is understandable and easy to read. The manuscript is generally well written and clear.
However, certain areas of the manuscript require further clarification to enhance its overall quality.
1. The study consists of 19 lung transplant recipients, representing a relatively small sample size. Although the statistical analyses address this limitation, the generalizability of the results remains uncertain. Authors should be more specific on how those 19 patients were selected. Were any exclusion criteria applied beyond what is mentioned? Was power analysis conducted to assess whether this sample size is adequate for detecting significant effects? Could the authors provide insight into how representative this cohort is in comparison to other lung transplant populations?
2. Authors should expand the discussion to include practical implications for clinical management such as if there were any unexpected adverse effects associated with tixagevimab/cilgavimab in this population? Was the infection rate or severity lower in this study than what has been reported in previous cohorts?
3. The study reports antibody binding titers (kBAU/ml) but does not assess neutralization efficacy. The study relies on binding titers alone, which do not always correlate with virus neutralization capacity—especially in immunosuppressed patients. Without neutralization efficacy data, it is difficult to conclude whether the observed antibody response is functionally protective. Authors should explicitly state that neutralization efficacy was not assessed and discuss why it was not included and reference other studies that have correlated binding antibody titers (kBAU/ml) with neutralization potential.
4. The use of Bonferroni correction is appropriate; however, the authors should clarify whether the correction was applied across all cytokine comparisons or separately for each time point. Consider discussing potential limitations of Bonferroni correction, such as increased Type II errors (false negatives), especially given the small sample size and multiple cytokine comparisons. Authors should consider False Discovery Rate (FDR) correction to recalculate p-values using FDR (Benjamini-Hochberg) correction for multiple testing instead of Bonferroni and compare results with Bonferroni-corrected values.
5. A key strength of this study is its cytokine profiling, particularly the insights into IL-18's role in protective immunity. Authors should include a comparison to prior studies on IL-18 in transplant recipients or COVID-19 PrEP patients and discuss whether IL-18 can be used as a biomarker for predicting PrEP efficacy.
The study associates higher IL-18 levels at T3 with protection against SARS-CoV-2 infection, but causation is not established. IL-18 plays a dual role in inflammation and does not always signify a beneficial immune response. Authors should discuss the possibility that IL-18 elevation is a secondary effect of T/C rather than a direct protective factor. The conclusions about immune protection should be more tentative.
Author Response
The manuscript presented for review titled “Immune Modulation and Efficacy of Tixagevimab/Cilgavimab Pre-Exposure Prophylaxis in Lung Transplant Recipients During the Omicron Wave" presents a study on the effects of tixagevimab/cilgavimab (T/C) pre-exposure prophylaxis (PrEP) in lung transplant recipients during the Omicron wave.
The subject matter of this article is important as it provides valuable clinical and immunological insights, particularly regarding cytokine responses and antibody dynamics. The systematic analysis of the topics raised by the Authors has been presented in a clear and coherent manner. The language of the work is understandable and easy to read. The manuscript is generally well written and clear.
However, certain areas of the manuscript require further clarification to enhance its overall quality.
- The study consists of 19 lung transplant recipients, representing a relatively small sample size. Although the statistical analyses address this limitation, the generalizability of the results remains uncertain. Authors should be more specific on how those 19 patients were selected. Were any exclusion criteria applied beyond what is mentioned? Was power analysis conducted to assess whether this sample size is adequate for detecting significant effects? Could the authors provide insight into how representative this cohort is in comparison to other lung transplant populations?
A1: Thank you for your observation. We acknowledge that the small sample size is a limitation of our study. However, it is important to note that the study was conducted during the COVID-19 pandemic, which significantly constrained the recruitment period. At that time, 149 lung transplant recipients were under regular follow-up at our center, and all were contacted by phone and invited to participate. Despite these efforts, only 19 patients were ultimately enrolled. Several exclusion criteria contributed to this outcome: 26 patients had experienced a recent SARS-CoV-2 infection (within the past 3 months), 23 patients already had detectable antibodies, 3 patients had a body weight below 40 kg, 10 patients had coagulation disorders for which anticoagulant therapy could not be safely discontinued, and 8 patients had previously received Tixagevimab/Cilgavimab therapy. Among those for whom the study treatment was proposed, 26 patients resided outside the Veneto region and declined to visit the study center due to concerns about traveling during the pandemic and the associated risk of SARS-CoV-2 infection. Additionally, 25 patients chose not to undergo the monoclonal antibody combination treatment, and 9 patients did not provide written informed consent.
To address these points clearly, we have revised the “Materials and Methods” section and included a study flowchart to better illustrate the selection criteria and provide a transparent overview of the study process.
In relation to the representative sample of our cohort compared to other lung transplant populations reported in the literature, we have added some clarifications to the discussion section at lines 210-219. “Although our cohort is small, it is comparable to those described in the literature regarding time since transplantation, immunosuppressive regimens, and underlying conditions” (Efficacy of pre-exposure prophylaxis to prevent SARS-CoV-2 infection after lung transplantation: a two center cohort study during the omicron era Infection. 2023 Oct;51(5):1481-1489. doi: 10.1007/s15010-023-02018).
Regarding the concerns on the power analysis, as previously mentioned, we are aware of the small sample size in our patient cohort. However, the primary aim of our study was to analyze and monitor the immune response and cytokine dynamics over time in lung transplant patients receiving Tixagevimab/Cilgavimab as PrEP. Our objective was not to assess the efficacy of this therapy, which would require a significantly larger number of patients recruited through a multicenter study, given the limited number of lung transplant recipients available in a single center. To note, we did not conduct a formal power analysis, as this was an exploratory, hypothesis-generating study aimed at characterizing immunological trends in a highly specific and clinically relevant population. Nonetheless, we applied rigorous statistical methods appropriate for small sample sizes, and we believe the results provide valuable preliminary insights that warrant further investigation in larger cohorts. A retrospective power analysis, considering the primary outcome variables, type of statistical tests conducted, significance level α = 0.05, number of groups, and a power 1-β = 0.80, suggests that our sample may be underpowered to detect small or medium effects.
- Authors should expand the discussion to include practical implications for clinical management such as if there were any unexpected adverse effects associated with in this population? Was the infection rate or severity lower in this study than what has been reported in previous cohorts?
A2: Thank you for the comment. Following the administration of monoclonal antibodies, severe hypersensitivity reactions, including anaphylaxis, were observed, but these were not seen in the patients enrolled in our study. Additionally, in the PROVENT registration study (N. Engl. J. Med. 2022, 386, 2188–2200), participants in the Tixagevimab/Cilgavimab arm experienced more serious cardiovascular adverse events compared to those in the placebo arm (0.7% vs 0.3%), particularly coronary events (e.g., myocardial infarction). A minor imbalance was observed for serious thromboembolic events (0.5% vs 0.2%). No cardiovascular events were observed during the clinical follow-up of the study. In the registration study, injection-related reactions were also commonly reported, including chills, and redness, discomfort, or pain near the injection site. These effects were not observed in the patients enrolled in the study.
We added these considerations in the discussion section (lines 213-218).
- The study reports antibody binding titers (kBAU/ml) but does not assess neutralization efficacy. The study relies on binding titers alone, which do not always correlate with virus neutralization capacity—especially in immunosuppressed patients. Without neutralization efficacy data, it is difficult to conclude whether the observed antibody response is functionally protective. Authors should explicitly state that neutralization efficacy was not assessed and discuss why it was not included and reference other studies that have correlated binding antibody titers (kBAU/ml) with neutralization potential.
A3: Thank you for your constructive comment. As correctly noted, our study reports anti-SARS-CoV-2 binding antibody titers (kBAU/ml) without assessing neutralizing efficacy. In this study, a commercially available immunoassay was used, the anti- SARS-CoV-2 receptor binding domain (RBD) IgG (Snibe Diagnostics, New Industries Biomedical Engineering Co., Ltd [Snibe], Shenzhen, China). SARS-CoV-2 S-RBD IgG is a chemiluminescent immunoassay (CLIA) that determines IgG Ab against the RBD of the Spike (S) protein of the virus, in human serum or plasma. All analyses were performed on MAGLUMI™2000Plus(SnibeDiagnostics), with results expressed in kBAU/ml. Several studies have demonstrated that the recombinant SARS-CoV-2 RBD is a highly sensitive and specific antigen for the detection of antibodies induced by SARS-CoV-2 and that the levels of RBD-binding antibodies positively correlates with neutralizing antibodies in COVID-19 patients (Front. Cell. Infect. Microbiol. , 14 April 2022, Sec. Clinical Microbiology,Volume 12 - 2022 | https://doi.org/10.3389/fcimb.2022.822599, https://doi.org/10.1016/j.heliyon.2024.e24513, https://doi.org/10.1038/s41541-022-00586-7). We specify in the “Material and Methods” section the immunoassay used (lines 311-316) and included in “discussion” section limitation and references of studies correlating SARS-CoV-2 S-RBD IgG titers with neutralization efficacy (lines 228-231).
- The use of Bonferroni correction is appropriate; however, the authors should clarify whether the correction was applied across all cytokine comparisons or separately for each time point. Consider discussing potential limitations of Bonferroni correction, such as increased Type II errors (false negatives), especially given the small sample size and multiple cytokine comparisons. Authors should consider False Discovery Rate (FDR) correction to recalculate p-values using FDR (Benjamini-Hochberg) correction for multiple testing instead of Bonferroni and compare results with Bonferroni-corrected values.
A4: Thank you for your suggestions. The Bonferroni correction was applied separately for each cytokine comparison. Since Bonferroni is a very conservative approach to controlling for multiple comparisons, and given the small number of patients and the large number of biomarkers, we believe that applying it separately for each cytokine comparison represents a reasonable tradeoff to minimize Type I errors without significantly reducing statistical power. We have clarified this in the Methods section (lines 365-3767).
- A key strength of this study is its cytokine profiling, particularly the insights into IL-18's role in protective immunity. Authors should include a comparison to prior studies on IL-18 in transplant recipients or COVID-19 PrEP patients and discuss whether IL-18 can be used as a biomarker for predicting PrEP efficacy.
The study associates higher IL-18 levels at T3 with protection against SARS-CoV-2 infection, but causation is not established. IL-18 plays a dual role in inflammation and does not always signify a beneficial immune response. Authors should discuss the possibility that IL-18 elevation is a secondary effect of T/C rather than a direct protective factor. The conclusions about immune protection should be more tentative.
A5: We appreciate the reviewer’s thoughtful comment regarding the role of IL-18 in our study. We agree that IL-18 plays a complex role in the immune response, with both protective and pro-inflammatory effects, and that caution is warranted in interpreting its elevation as a causal marker of protection. While we acknowledge the importance of contextualizing our findings, we believe that comparisons with studies on transplant patients outside the context of SARS-CoV-2 pre-exposure prophylaxis (PrEP) may not be fully informative. Our study does not aim to evaluate transplant success or immune reconstitution per se, but rather to investigate immune modulation and protection conferred by tixagevimab/cilgavimab in a PrEP setting during the Omicron wave. To our knowledge, this is the first study to report longitudinal cytokine profiling, including IL-18, in lung transplant recipients undergoing PrEP with tixagevimab/cilgavimab. While there is limited literature specifically linking IL-18 to PrEP efficacy, prior studies have associated low IL-18 levels with poorer overall survival in COVID-19 patients in general populations (Sun et al., Signal Transduct Target Ther, 2023) [https://www.nature.com/articles/s41392-023-01368-w]. Conversely, our study by Angioni et al. (Cell Death Dis, 2020) [https://www.ncbi.nlm.nih.gov/pmc/articles/PMC7646225/] did not find IL-18 to be significantly associated with COVID-19 severity, reflecting the heterogeneity of IL-18's clinical implications. Importantly, in our cohort, IL-18 was the only cytokine significantly upregulated at T3 in patients who did not develop COVID-19 after PrEP. However, we agree that causality cannot be established, and that IL-18 elevation may reflect an indirect effect of immune system priming triggered by tixagevimab/cilgavimab, rather than a direct protective factor. We have revised the discussion (lines 258-262) to reflect a more cautious interpretation of IL-18’s role and highlight the need for further studies to clarify its utility as a predictive biomarker of PrEP efficacy.

Reviewer 2 Report
Comments and Suggestions for Authors
Sasset and colleagues submit a manuscript regarding immune modulation and efficacy of tixagevimab/cilgavimab pre-exposure prophylaxis in 19 lung transplant recipients during the Omicron wave. They are addressing the question as to whether monoclonal antibody use can prevent infection with the SARS-CoV-2 virus. The topic is relevant to the field, as it adds more data to the growing body of knowledge regarding monoclonal antibody use for this infection. The methods section is placed at the end, which is distracting to this reviewer, but that may be a policy of this particular journal. The methods themselves appear to be fine, other than what is listed in comments below. Their conclusion that this monoclonal antibody probably has protective effects is supported by the data in their results section, although the authors could do a better job of explaining some of their data for the reader who is not as technically advanced with their methods. Their conclusion that the cases of breakthrough infection were mild as a result of preventative use of this monoclonal antibody is reasonable.
Comments:
- Based on the title, the study took place during the Omicron wave. The Omicron wave is generally considered to be between November 2021 through to February 2022. In the methods, the authors report that Tixagevimab/cilgavimab was administered by intramuscular injections at the dose of 300 mg according to clinical indication between May and June 2022, when Omicron SARS-CoV-2 variant was prevalent in Italy. Please explain this apparent discrepancy between what I as a reader would assume regarding the Omicron wave, and when the monoclonal antibody was actually given.
- Please list start and stop dates for the study. Was this a study that started on May 1, 2022, and ended on June 30, 2022?
- Lung transplant is not a high-volume procedure, so it is understandable that the sample size is limited to 19 patients. Perhaps the authors could state whether this was their entire volume of transplanted patients during the study period. If this is not their entire volume, then what were reasons that some transplant recipients were not included? Should a consort diagram be provided regarding reasons for exclusion from this study?
- Since this is such a small-scale analysis of data (19 patients), do the authors feel that the increase in IL-18, highlighted in their discussion as being more pronounced for those transplant recipients on immunosuppression for more than 5 years, is a true finding? Table 1 lists then time after transplant as being a range of 3 to 9 years. Of the 19 patients, how many were in the 3 to 5 year range, and how many were in the 5 to 9 year range? Also, is the explanation of IL-18 good enough in the figure legends for Figures 2 and 3 to follow the data, particularly for the novice reader?
- Should any of the biomarkers in the heatmap of Panel A in Figure 3 be used to follow patients clinically?
- Why has this study taken 3 years to come together as a manuscript?
- In the sentence “Five patients (83%) developed a COVID-19…”, is the word infection missing after COVID-19?
- In the second paragraph of the methods section, there is a long reference in parentheses. Shouldn’t this have been picked up by the reference manager and listed in the bibliography by a number?
Author Response
REVIEWER 2
Sasset and colleagues submit a manuscript regarding immune modulation and efficacy of tixagevimab/cilgavimab pre-exposure prophylaxis in 19 lung transplant recipients during the Omicron wave. They are addressing the question as to whether monoclonal antibody use can prevent infection with the SARS-CoV-2 virus. The topic is relevant to the field, as it adds more data to the growing body of knowledge regarding monoclonal antibody use for this infection. The methods section is placed at the end, which is distracting to this reviewer, but that may be a policy of this particular journal. The methods themselves appear to be fine, other than what is listed in comments below. Their conclusion that this monoclonal antibody probably has protective effects is supported by the data in their results section, although the authors could do a better job of explaining some of their data for the reader who is not as technically advanced with their methods. Their conclusion that the cases of breakthrough infection were mild as a result of preventative use of this monoclonal antibody is reasonable.
Comments:
- Based on the title, the study took place during the Omicron wave. The Omicron wave is generally considered to be between November 2021 through to February 2022. In the methods, the authors report that Tixagevimab/cilgavimab was administered by intramuscular injections at the dose of 300 mg according to clinical indication between May and June 2022, when Omicron SARS-CoV-2 variant was prevalent in Italy. Please explain this apparent discrepancy between what I as a reader would assume regarding the Omicron wave, and when the monoclonal antibody was actually given.
A1: Thank you for giving us the opportunity to clarify this point. Based on the report from the Istituto Superiore di Sanità (Prevalence and Distribution of SARS-CoV-2 Variants of Concern for Public Health in Italy, Report No. 22 of July 28, 2022, (data updated as of July 21, 2022) the Omicron variant was the only circulating SARS-CoV-2 variant in our country. The Omicron variant remained the only circulating variant in Italy, with an estimated national prevalence of 100% in July 2022.
- Please list start and stop dates for the study. Was this a study that started on May 1, 2022, and ended on June 30, 2022?
A2: The study enrolled 19 lung transplant patients who consented to participate in the survey from May 1 to June 30, 2022. Enrolled patients were clinically assessed for 6 months after the administration of the Evusheld drug. The study therefore ended on January 30, 2023, at the conclusion of the clinical follow-up of the last enrolled patient. We have added these details in the Methods Section (Lines 295-297).
- Lung transplant is not a high-volume procedure, so it is understandable that the sample size is limited to 19 patients. Perhaps the authors could state whether this was their entire volume of transplanted patients during the study period. If this is not their entire volume, then what were reasons that some transplant recipients were not included? Should a consort diagram be provided regarding reasons for exclusion from this study?
A3: Thank you for your observation. We acknowledge that the small sample size is a limitation of our study. However, it is important to note that the study was conducted during the COVID-19 pandemic, which significantly constrained the recruitment period. At that time, 149 lung transplant recipients were under regular follow-up at our center, and all were contacted by phone and invited to participate. Despite these efforts, only 19 patients were ultimately enrolled. Several exclusion criteria contributed to this outcome: 26 patients had experienced a recent SARS-CoV-2 infection (within the past 3 months), 23 patients already had detectable antibodies, 3 patients had a body weight below 40 kg, 10 patients had coagulation disorders for which anticoagulant therapy could not be safely discontinued, and 8 patients had previously received Tixagevimab/Cilgavimab therapy. Among those for whom the study treatment was proposed, 26 patients resided outside the Veneto region and declined to visit the study center due to concerns about traveling during the pandemic and the associated risk of SARS-CoV-2 infection. Additionally, 25 patients chose not to undergo the monoclonal antibody combination treatment, and 9 patients did not provide written informed consent.
To address these points clearly, we have revised the “Materials and Methods” section and included a study flowchart to better illustrate the selection criteria and provide a transparent overview of the study process.
- Since this is such a small-scale analysis of data (19 patients), do the authors feel that the increase in IL-18, highlighted in their discussion as being more pronounced for those transplant recipients on immunosuppression for more than 5 years, is a true finding? Table 1 lists then time after transplant as being a range of 3 to 9 years. Of the 19 patients, how many were in the 3 to 5 year range, and how many were in the 5 to 9 year range? Also, is the explanation of IL-18 good enough in the figure legends for Figures 2 and 3 to follow the data, particularly for the novice reader?
A4: We thank the reviewer for these thoughtful and constructive comments. We fully acknowledge that the small sample size represents a limitation and that any subgroup analysis—such as the one exploring IL-18 trends in relation to immunosuppression duration—should be interpreted with caution. Our aim was not to make definitive claims, but rather to report an observation that may be hypothesis-generating for future, larger studies.
To clarify the reviewer’s question: among the 19 patients, 8 had been receiving immunosuppressive therapy for more than 5 years (range: 6–15 years), 8 for 2–5 years, and 3 patients had been on immunosuppressive therapy for less than 1 year. We have now included this detail in the revised text for transparency (Lines 89-91). Regarding the figure legends, we intentionally kept the figure legends concise to maintain clarity and avoid redundancy with the main text. While the legends for Figures 2 and 3 briefly mention IL-18, a more detailed explanation of its immunological relevance and role in our findings is provided in the Results and Discussion sections. We aimed to strike a balance between readability and completeness, assuming that readers seeking deeper insight would refer to the main text. However, we are happy to expand the legends if the journal prefers a more descriptive approach, particularly for the benefit of less-experienced readers.
- Should any of the biomarkers in the heatmap of Panel A in Figure 3 be used to follow patients clinically?
A5: We thank the reviewer for raising this important point. Indeed, the ultimate goal of this type of immunomonitoring is to identify reliable biomarkers that could be used clinically to guide patient follow-up and risk stratification. However, this was beyond the scope of the present study for several reasons. Importantly, the heterogeneity of immunosuppressive regimens and clinical histories among lung transplant recipients introduces variability that would require a larger cohort to control for. Thus, we are planning to further recruit patients for data validation and eventually to move toward clinical practice. Nevertheless, our results provide a starting point and rationale for future prospective studies aimed at validating selected biomarkers—such as IL-18 or TNF-α—as part of personalized follow-up strategies in immunocompromised populations.
- Why has this study taken 3 years to come together as a manuscript?
A6: We thank the reviewer for this observation. The study was conducted during a critical phase of the pandemic, and patient follow-up extended over a 6-month period. Additional time was required for cytokine profiling, data analysis, and interpretation, particularly given the complexity of immunological assessments in immunosuppressed individuals. Furthermore, all authors are actively engaged in clinical duties and academic responsibilities, alongside participation in other ongoing scientific projects, which contributed to the extended timeline. Nevertheless, we made every effort to ensure that the manuscript presents a thorough and carefully contextualized analysis within the evolving landscape of SARS-CoV-2 prophylaxis.
- In the sentence “Five patients (83%) developed a COVID-19…”, is the word infection missing after COVID-19?
A7: We thank the reviewer for the observation. However, no modification was made, as the sentence is correctly formulated. In this context, we specifically refer to the development of COVID-19 as the clinical disease resulting from SARS-CoV-2 infection. The term “infection” is not missing, as the phrase “developed a COVID-19” is meant to describe the symptomatic condition, consistent with standard usage in clinical literature and guidelines.
- In the second paragraph of the methods section, there is a long reference in parentheses. Shouldn’t this have been picked up by the reference manager and listed in the bibliography by a number?
A8: We thank the reviewer for noticing this formatting inconsistency. You are absolutely right — this reference should have been handled by the reference manager and listed numerically. We have now corrected it and included the citation properly in the reference list.

Round 2
Reviewer 1 Report
Comments and Suggestions for Authors
I would like to thank the authors for their thoughtful and comprehensive revisions. The concerns raised in the previous review have been addressed with clarity and scientific rigor. The updated manuscript is well-structured, the methods and discussion have been appropriately expanded, and the authors have demonstrated a strong understanding of the study’s limitations.
I am satisfied with the revised version and have no further suggestions